# The Epithelial to Mesenchymal Transition in Colorectal Cancer Progression: The Emerging Role of Succinate Dehydrogenase Alterations and Succinate Accumulation

**DOI:** 10.3390/biomedicines11051428

**Published:** 2023-05-11

**Authors:** Mimmo Turano, Rosario Vicidomini, Francesca Cammarota, Valeria D’Agostino, Francesca Duraturo, Paola Izzo, Marina De Rosa

**Affiliations:** 1Department of Biology, University of Naples Federico II, 80126 Naples, Italy; mimmo.turano@unina.it; 2Section on Cellular Communication, Eunice Kennedy Shriver National Institute of Child Health and Human Development, National Institutes of Health, Bethesda, MD 20892, USA; rosario.vicidomini@nih.gov; 3Department of Molecular Medicine and Medical Biotechnology, University of Naples Federico II, 80131 Naples, Italy; 4CEINGE-Biotecnologie Avanzate Franco Salvatore, 80131 Naples, Italy

**Keywords:** colorectal cancers (CRCs), epithelial to mesenchymal transition (EMT), succinate dehydrogenase (SDH), succinate, metabolic reprogramming

## Abstract

Colorectal cancer (CRC) stands as the third most significant contributor to cancer-related mortality worldwide. A major underlying reason is that the detection of CRC usually occurs at an advanced metastatic stage, rendering therapies ineffective. In the progression from the in situ neoplasia stage to the advanced metastatic stage, a critical molecular mechanism involved is the epithelial-to-mesenchymal transition (EMT). This intricate transformation consists of a series of molecular changes, ultimately leading the epithelial cell to relinquish its features and acquire mesenchymal and stem-like cell characteristics. The EMT regulation involves several factors, such as transcription factors, cytokines, micro RNAs and long noncoding RNAs. Nevertheless, recent studies have illuminated an emerging link between metabolic alterations and EMT in various types of cancers, including colorectal cancers. In this review, we delved into the pivotal role played by EMT during CRC progression, with a focus on highlighting the relationship between the alterations of the tricarboxylic acid cycle, specifically those involving the succinate dehydrogenase enzyme, and the activation of the EMT program. In fact, emerging evidence supports the idea that elucidating the metabolic modifications that can either induce or inhibit tumor progression could be of immense significance for shaping new therapeutic approaches and preventative measures. We conclude that an extensive effort must be directed towards research for the standardization of drugs that specifically target proteins such as SDH and SUCNR1, but also TRAP1, PDH, ERK1/2, STAT3 and the HIF1-α catabolism.

## 1. Introduction

Colorectal cancer (CRC), constituting approximately 10% of the 19 million newly diagnosed cancer cases worldwide, is the third most common cancer, ranking behind breast cancer in women and lung cancer. As reported by GLOBOCAN cancer statistics 2020 [1], CRC is the second most lethal cancer, following lung cancer, accounting for about 9.4% of the roughly 10 million cancer-related fatalities. 

Most colorectal cancers develop according to a well-known adenoma–carcinoma sequence, during which the normal epithelium progressively accumulates genetic and epigenetic alterations, which confer onto the cells a selective advantage in growth. Consequently, the once normal mucosa transforms into hyperproliferative mucosa and, subsequently, into early, intermediate and late adenoma, ultimately leading to a carcinoma. If not promptly removed, in situ carcinoma often evolves into a metastatic carcinoma that is capable of moving from the site of the primary lesion and colonizing at a distance from organs and tissues [2]. 

Based on cancer location and size, it grows into nearby tissues, and spreads to nearby lymph nodes or other parts of the body; the disease can be classified into four stages, where stages I and II are characterized by in situ tumors, while stages III and IV correspond to local lymph node invasion and distant metastases. Although surgery represents a viable treatment option in the first two stages of the disease, it is often ineffective for stage III and IV, which often exhibit resistance to currently available systemic therapies [3,4].

During colorectal cancer progression, tumor cells undergo EMT by losing their apico-basal polarity and cell adhesion and acquiring the typical pluripotency of mesenchymal cells [5]. 

Different studies have highlighted a close connection between metabolic alterations and EMT induction during tumor progression. Glucose, lipid and amino acid metabolism alterations, in fact, significantly contribute to inducing EMT and promoting the invasion of tumor cells [6,7]. An important role, in this context, is played by tricarboxylic acid (TCA) cycle enzymes, such as succinate dehydrogenase (SDH), whose dysfunction is associated with EMT-driven tumorigenesis [8,9,10,11,12,13].

This review is focused on the most recent advances regarding the role of the succinate dehydrogenase enzyme in CRC. We first provide an overview on CRCs and their classification, addressing the signaling pathways involved in EMT activation. Then, we explore how succinate dehydrogenase alterations and succinate accumulation promote EMT and tumor progression through both intrinsic and extrinsic pathways.

## 2. CRCs: Epidemiology and Genetics

The majority of colorectal malignant neoplasms fall into the category of adenocarcinomas, a type of tumor that arises from epithelial cells. However, there exists a rare subset of CRCs that make up 0.1% of cases, known as gastrointestinal stromal tumors (GISTs), which refers to gastrointestinal tumors of mesenchymal type that originated from Cajal cells [14].

Approximately 70% of colon tumors are classified as sporadic, arising from somatic alterations, and only 10% have Mendelian inheritance, which is caused by germline pathogenetic alterations in genes essential for the correct turnover of epithelial cells within the colorectal mucosa. The remaining cases are instead familial in nature, exhibiting a familial predisposition to the development of CRC without a Mendelian inheritance pattern. It is hypothesized that these cases may arise due to alterations in genes with incomplete penetrance and that they often manifest phenotypic heterogeneity [2,3,4].

The collection of hereditary syndromes that increase individuals’ susceptibility to colorectal cancer is extensive and are both phenotypically and genetically heterogeneous. They can be broadly categorized as familial gastrointestinal polyposis and hereditary non-polyposis colorectal cancer (HNPCC) syndromes. Polyposis syndromes are characterized by the onset of polyps, in a variable number, along the gastrointestinal tract. HNPCC or Lynch syndrome, is marked by the predisposition to colorectal cancer and to other extraintestinal tumors, such as endometrium, ovary, stomach, small intestine, hepatobiliary tract, pancreas, urinary tract, prostate, brain and skin, commonly associated with the syndrome [15,16]. This increased susceptibility to tumors is observed in the absence of gastrointestinal polyps or with the presence of a limited number of them. Moreover, based on the histological polyp features (adenomatous or hamartomatous), polyposis syndromes are divided into two other distinct groups: familial adenomatous polyposis (FAP) and hamartomatous polyposis. In addition, two other groups of syndromes namely mixed polyposis and serrated polyposis, are part of the broad family of polyposis syndromes [3]. At the genetic level, these syndromes are incredibly heterogeneous. Table 1 highlights the main hereditary syndromes that increase the likelihood of developing colorectal cancer and the corresponding disease genes involved. Many germline pathogenic variants in each of these genes cause the diseases’ onset, with penetrance varying according to the specific syndrome [17,18,19,20]. We demonstrated that quantitative alterations in the expression of the same genes contribute to generate inter- and intra-familial phenotypic heterogeneity [21,22]. Furthermore, germline pathogenic variants in specific gene, such as the phosphatase and tensin 106 homolog (PTEN) gene, generate constitutive beta-catenin and cytokine dysregulation, activating inflammatory pathways and, in turn, increasing cancer risk of carrier subjects [23]. 

Importantly, about 10% of subjects affected by Cowden syndrome (CS) or Cowden-like syndrome (CLS), negative for pathogenic variants of the *PTEN* gene, are carriers of a germline pathogenic variant in one of the genes coding for SDH [24]. Furthermore, alterations of *SDH* genes have been described in various human diseases, including hereditary and sporadic tumors such as gastrointestinal stromal tumors [25,26,27], pheochromocytoma, paraganglioma, renal carcinomas and papillary thyroid carcinomas [28,29,30,31,32]. The ClinVar database (https://www.ncbi.nlm.nih.gov/clinvar/, accessed on 1 March 2023), reports *SDH* variants in several individuals affected by gastrointestinal tumors. To date, seven variants of “uncertain significance/conflicting interpretations” of pathogenicity and three “pathogenic/likely” variants, have been described in the *SDHA* gene; all variants have been associated with patients affected by GIST (ORPHA:44890) and/or hereditary cancer-predisposing syndrome (ORPHA:140162). Additionally, nine variants of uncertain significance/conflicting interpretations of pathogenicity and seventeen pathogenic/likely variants, have been described in the *SDHB* gene; all variants have been associated with patients affected by GIST and/or hereditary cancer-predisposing syndrome; only one variant has been associated with a patient affected by CS (ORPHA:201). Furthermore, three variants of uncertain significance/conflicting interpretations of pathogenicity have been described in the *SDHC* gene; all variants have been associated with patients affected by GIST and/or hereditary cancer-predisposing syndrome. Finally, six variants of uncertain significance/conflicting interpretations of pathogenicity and six pathogenic/likely variants, have been described in the *SDHD* gene; all variants have been associated with patients affected by hereditary cancer-predisposing syndrome, while eleven variants have been associated with patients affected by CS.

Chromosome instability (CIN), alterations in methylation that generate the CpG island methylator phenotype (CIMP), and the alteration of DNA repair systems that results in microsatellite instability (MSI), are all mechanisms that cause genetic variability in colon epithelial cells during their tumor transformation. These mechanisms underlie the unique characteristics of each tumor and determine its responsiveness to specific therapeutic interventions [3,4]. 

Given the genetic heterogeneity of colorectal cancers, Guinney and co-workers [33] set up a consortium with the primary aim of sharing data obtained by several expert groups of scientists. By integrating six distinct classifications, they were able to identify four cancer molecular subtypes (CMS), as reported in the graph of Figure 1. This classification, probably the most comprehensive currently existing for colorectal cancers, has great translational potential for the therapy and management of these patients. The authors also identified a group, representative of about 13% of all samples, showing mixed features that could represent transitional phenotypes from one subgroup to another. However, findings derived from single-cell RNA sequencing analysis indicate that many CRCs may have mixed subtypes and intratumor genetic variability [34]. More recently, Joanito and coworkers, in a single-cell RNA sequencing study, identified additional subgroups, including intrinsic-CMS3, characterized by association of EMT, inflammatory pathways and metabolic rearrangements [35].

## 3. Role of the EMT in Colon Cancer Progression

EMT is a physiological process characterized by epithelial cells losing their specific characteristics to acquire those of a mesenchymal cell. While this process is mainly involved during embryogenesis and development, it can also be activated in adulthood, under specific conditions such as wound healing processes and tumor progression of various malignancies including colorectal tumors. During colorectal cancer progression, tumor cells lose their epithelial characteristics, including apical–basal cell polarity and the expression of membrane proteins involved in cell–cell tight junctions, such as E-cadherins, a pivotal gatekeeper protein of the epithelial tissue. Concurrently, tumor cells increase the expression of proteins specific to a mesenchymal phenotype, such as N-cadherin and vimentin; consequently, they acquire motility and a stem-like phenotype, showing high expression of stem-cell markers, self-renewal capability, pluripotency and resistance to apoptosis [36,37]. Due to all these features, it has been hypothesized that EMT represents a mechanism that provides tumor cells with a pool of stem-like cells that promote tumor progression towards metastases formation. Indeed, tumor cells that adopt a mesenchymal phenotype can degrade the membrane basement of the epithelial tissue, invade the surrounding tissues, and pass through the endothelium into the lumen of the vessels; a process defined as intravasation. Subsequently, within the lumen of the vessels, these cells, resistant to the adverse environment, can exit from the vessels, with a process called extravasation, and finally colonize at distant organs. Conversely, the reversal process of EMT, mesenchymal-to-epithelial transition (MET), is instead implicated in the colonization of remote organs by the tumor cell, thereby driving the metastasis formation. 

We have clearly demonstrated that, by reversing the mesenchymal phenotype, colon cancer cells lose their stem-cell-like features, are able to differentiate, reduce their ability to migrate and invade surrounding tissues and show a better response to therapeutic treatments, such as photon irradiation [38,39].

## 4. Signaling Pathways Involved in EMT Activation during Colon Cancer Progression

The intricate process of EMT, involved in colorectal cancer progression, is governed by numerous transcription factors and signaling pathways. Notably, among the main regulators are specific transcription factor proteins, called EMT-TFs, including twist-related protein 1 (TWIST1), twist-related protein 2 (TWIST2), zinc finger protein SNAI1 (SNAI1), zinc finger protein SNAI2 (SNAI2), zinc finger E-box-binding homeobox 1 (ZEB1) and zinc finger E-box-binding homeobox 2 (ZEB2) [40,41]. They act by inhibiting the transcription of *E-Cadherin* (*CDH1*), while upregulating mesenchymal- and stem-cell-specific markers to induce the transition towards a mesenchymal phenotype. 

SNAI1 acts by repressing the *E-cadherin* expression and interacting with the factors involved in chromatin remodeling [42]. Additionally, it interacts with the apical transmembrane protein crumbs homolog 3 to alter the characteristic polarity of the epithelial cells [41,43]. Finally, it promotes cell survival by repressing programmed cell death via apoptosis and by blocking the cell cycle [44]. We demonstrated that *SNAI2* is upregulated in colorectal cancers, and correlates with a cancer-stem-like cell phenotype and the expression of specific stem cell markers [37,38].

Other transcription factor proteins that regulate the EMT process during CRC progression belong to the family of basic helix-loop-helix transcription factors (bHLH-TFs), which include TWIST1, TWIST2, transcription factor 3 (TCF3) and transcription factor 4 (TCF4) [45,46]. Similar to SNAI, TWIST proteins also repress the expression of *CDH1* and simultaneously activate *N-cadherin* (*CDH2*), *vimentin* (*VIM*) and *fibronectin 1* (*FN1*) expression, which in turn induce a shift from the epithelial phenotype towards a mesenchymal one [45,47]. The transcription factor TWIST1 is upregulated during embryonic morphogenesis, wound healing and fibrotic processes [48,49]. Conversely, it is poorly expressed in differentiated adult cells in physiological conditions. However, its overexpression has been observed in tumor transformation and is closely associated with metastatic progression and a poor prognosis of the disease [50].

ZEB1 and ZEB2, both zinc-finger proteins, function as transcription factors [51]. They repress epithelial markers, including *CDH1*, *LLGL scribble cell polarity complex component 2* (*LLGL2*), *PATJ crumbs cell polarity complex component* (*PATJ*) and *crumbs cell polarity complex component 3* (*CRB3*) [52,53] and activate mesenchymal and stem-like markers [54]. Under physiological conditions, they are mainly expressed in the central nervous system, skeletal and cardiac muscle and hematopoietic cells [55]. More recently, the role of ZEB1 in chromatin remodeling has been demonstrated in tumors belonging to the colon mesenchymal subtype CMS4, which exhibits cyclin-dependent kinase inhibitor 1 (CDN1A, P21) alterations mediated by an up regulation of the histone-lysine N-methyltransferase SETD1B protein [56]. Moreover, ZEB2 expression has been found to be associated with poor prognosis in patients affected by CRC and with the progression of the disease towards metastases formation. These findings have significant clinical implications, as the use of ZEB1 as a prognostic marker, coupled with the TNM classification, can aid in the stratification of patients according to the risk of disease recurrence [55,56,57,58]. 

Another transcription factor that regulates the EMT is the paired-related homeobox protein 1 (PRRX1), which activates EMT in various types of cancers, including colorectal cancers [59]. Two isoforms of PRRX1, namely PRRX1a and PRRX1b, play diverse roles in EMT and tumor progression. Indeed, PRRX1a induces EMT, until a specific time in the progression of an in situ carcinoma, towards a metastatic disease. On the contrary, PRRX1b promotes the MET process and the formation of epithelial cells with stem-like characteristics that allow metastatic colonization [59,60]. 

A clinical study, conducted on a cohort of 29 CRC patients, showed a decrease in SDHB levels in affected subjects. This phenomenon is associated with the overexpression of mothers against decapentaplegic homolog 4 (SMAD4), SNAI1 and N-cadherin proteins, as well as the downregulation of E-cadherin. In vitro experiments conducted by the same authors demonstrated that the knockdown of *SDHB* in commercial SW1116 colon cancer cells led to the activation of the TGFβ signal pathway. This activation was enabled by the SNAI1-SMAD3/4 complex, which repressed the expression of E-cadherin. These cells exhibited overexpression of SNAI1, SMAD3, SMAD4, fibronectin and N-cadherin proteins, while the epithelial markers E-cadherin and tight junction ZO-1 proteins were downregulated [61]. Consequently, SDH appears to be a key enzyme regulating CRC aggressiveness, linking energy metabolism with biological processes that are typical of tumor progression, such as EMT, invasion and metastasis formation.

## 5. Role of Succinate Dehydrogenase Alterations and Succinate Accumulation in CRC Onset and Progression

SDH is a mitochondrial enzyme that physically connects the TCA cycle and the mitochondrial electron transport chain (ETC). It is a tetrameric enzyme composed of four different subunits encoded by four different genes: *succinate dehydrogenase complex flavoprotein subunit A* (*SDHA*), *succinate dehydrogenase complex iron sulfur subunit B* (*SDHB*), *succinate dehydrogenase complex subunit C* (*SDHC*) and *succinate dehydrogenase complex subunit D* (*SDHD*). The A and B subunits protrude towards the mitochondrial matrix, while the C and D subunits are transmembrane polypeptides of the inner mitochondrial membrane that help stabilize the tetramer (Figure 2). The A subunit of SDH contains a FAD binding region and forms a complex with the B subunit, which is an FeS protein [62,63]. Meanwhile, subunits C and D have a binding site for ubiquinone and its reduced form, ubiquinol. They also house heme-b, which provides structural support for the membrane anchoring enzyme region, rather than playing a main role in catalysis [62]. In a study conducted by Wang and co-workers, a decreased expression of *SDHB* subunit was observed in CRC patients and in commercially available colon cancer cell lines. They showed that alterations in the genes encoding SDH subunits cause enzyme dysfunction and succinate accumulation. The authors also demonstrated that *SDHB* knockdown induces an increase in colon cancer cell invasion and activation of the TGF-beta signaling pathway [61]. Furthermore, a decrease in *SDH* expression is associated with increased cell growth and de-differentiation of colon cancer cells, which is mediated by a decrease in expression of the tumor suppressor *PTEN* [64]. At the metabolic level, SDH functions simultaneously in both the TCA cycle and ETC level. In the TCA cycle, it catalyzes the succinate to fumarate oxidation through a FAD dependent reaction. The two electrons released by the substrate succinate are first transferred to the iron–sulfur (Fe-S) clusters housing into the complex, and subsequently to the ubiquinone (Q) which is reduced to ubiquinol (QH2) [65] (Figure 2). 

Pathogenetic variants in the genes encoding the succinate dehydrogenase subunits, alteration of their expression or inhibition of the SDH enzyme leads to the accumulation of succinate and a decrease in the fumarate concentration. Importantly, this process is also accompanied by the accumulation of reactive oxygen species (ROS) and alterations in the oxidative phosphorylation process. Notably, one of the known SDH inhibitors that determines the increase in succinate concentration is the TNF receptor-associated protein 1 (TRAP1), a molecular chaperone member of the heat shock protein family [66,67,68,69,70,71,72] (Figure 2).

Recent research has revealed that some microRNAs regulate the expression of metabolic genes, which induces a metabolic reprogramming during the tumor transformation of colorectal cancers and their progression towards a metastatic phenotype [73]. Specifically, one such microRNA, Mir-142-5p, is found to be overexpressed in colon tumor tissues. It inhibits the expression of the gene encoding for the *SDHB* subunit, which in turn activates aerobic glycolysis and accelerates glucose consumption and pyruvate production [74]. Finally, hypoxia-induced Mir-210 has been also shown to promote self-renewal in colorectal-cancer-initiating cells, through down-regulation of the *SDHD* subunit [75,76,77].

## 6. Role of Succinate in EMT, Angiogenesis and Inflammatory Pathways in CRC Cells

Succinate, apart from its metabolic role, has been shown to play a crucial role in the activation of inflammatory signaling pathways and the onset and progression of tumors [78,79,80]. The interaction of succinate with succinate receptor 1 (SUCNR1), also known as G protein-coupled receptor-91 (GPR91), on the surface of the epithelial colon cancer cells has been reported to activate EMT. Despite limited published studies elucidating the underlying mechanism, recent in vitro experiments have revealed that treating HT29 colorectal cancer cell cultures with succinate results in the upregulation of the SUCNR1 receptor, as well as in the upregulation of mesenchymal markers such as SNAI1, SNAI2 and vimentin. Additionally, succinate treatment leads to the downregulation of E -cadherin. 

The authors of the study also demonstrated that the binding of succinate to the SUCNR1 receptor induces the activation of the WNT pathway, upregulation of beta-catenin, its translocation to the nucleus and activation of downstream targets, such as *MYC proto-oncogene*, *leucine-rich repeat-containing G protein-coupled receptor 5* (*LGR5*), *JUN proto-oncogene*, *mitogen-activated protein kinase 9* and *cyclinD* [81]. 

In a recent study, the SUCNR1 receptor signaling pathway was investigated in human and murine lung cancer cell lines. The researchers demonstrated that the succinate-mediated activation of SUCNR1 induces HIF1-*α* stabilization via the PI-3k/AKT pathway and metastasis formation. They also found that the succinate-SUCNR1 interaction increases the intracellular calcium concentration [82]. These findings align with the nature of SUCNR1, which belongs to the family of G protein-coupled receptors that activate second messengers, such as calcium ions or cyclic AMP [83].

Furthermore, the interaction of succinate with SUCNR1 on the plasma membrane surface of macrophage, activates the expression of pro-inflammatory cytokine interleukin 1 beta (IL-1β) and interleukin 6 (IL-6), resulting in the stabilization of hypoxia-inducible factor 1-alpha (HIF1-α) and the activation of an inflammatory pathway (Figure 3) [84,85].

It has also been established that both IL-1β and IL-6 are overexpressed in CRCs, in agreement with the well-known role that the chronic inflammation process plays in supporting angiogenesis, tumor proliferation and migration, thus facilitating the transformation of carcinoma towards the metastases [86,87,88]. Furthermore, it has been demonstrated that succinate induces the angiogenesis process by activating the ERK/STAT3 signaling pathways [89] (Figure 3), and the epigenetic alterations by inactivating DNA and histone demethylases [90,91,92,93]. It is important to note that the receptor-mediated activation of specific signaling pathways is a context-dependent mechanism, varying across different cell types and tissues. Furthermore, these mechanisms can be influenced by the metabolic conditions. In light of the current understanding of the signaling pathways activated by the SUCNR1 receptor, we strongly advocate for further studies aimed at elucidating and demonstrating the precise signaling pathways activated in different tissue and cell types.

It has been known for a long time that tumor cells undergo metabolic remodeling, which shifts energy metabolism towards glycolysis acceleration even in the presence of oxygen, a phenomenon known as the Warburg effect [94,95,96]. The activation of glycolytic metabolism is then associated with the tumor suppressor’s inactivation and oncogene’s activation, which facilitate tumor onset and progression [97]. A central aspect in this mechanism is TCA cycle alterations, which occur during tumor transformation, predominantly via mutation of the SDH and the fumarate hydratase, leading to the accumulation of fumarate and succinate. Succinate, accumulated in the mitochondria, translocates into the cytoplasm, where it inhibits several enzymes with 2oxoKetoglutarate-dependent oxygenase activity, including prolyl hydroxylases (PHDs) and the TET (ten-eleven translocation) family of 5-methlycytosine (5mC) hydroxylases (TETs) (Figure 2). PHDs inactivation, in turn, causes EMT, pseudohypoxia and glycolysis activation (Figure 2 and Figure 3) [98,99]. The TET enzymes, on the other hand, catalyze the 5-methyl-cytosine oxidative demethylation. Their inhibition has been described in many cancers, including colorectal cancers, and causes subsequent DNA hypermethylation. As a result, succinate has been defined as an “epigenetic acker” [90,100] (Figure 2).

The master regulator of the response to oxygen deficiency, which can occur in physiological or pathological conditions including cancer, is the transcription factor HIF1 [101]. HIF1 is a heterodimer consisting of a catalytic and regulatory alpha subunit (HIF1-α) that is degraded in the presence of oxygen and a ubiquitously expressed beta subunit (HIF1-β), also termed aryl hydrocarbon receptor nuclear translocator (ARNT) [102]. Under optimal oxygenation conditions, the HIF1-α concentration generally remains low inside the cell due to two main post-translational modifications. First, the protein is hydroxylated at proline residues, under a reaction catalyzed by PHD enzymes [103]. Subsequently, the hydroxylated protein binds to the von Hippel–Lindau protein (VHL) and is ubiquitinated and targeted for proteasomal degradation [104]. However, under hypoxia conditions or PHD inhibition, HIF1-α stabilizes and translocates into the nucleus where it dimerizes with the HIF-1beta subunit and generates the active HIF transcription factor. The active HIF transcription factor, in turn, activates the transcription of specific genes, including those involved in the activation of glycolysis and angiogenesis. Cancer cells often exhibit pseudohypoxia, the activation of the hypoxia pathway under normal oxygenation conditions. This process is caused by the overexpression of HIF1-α, mediated by the mTOR pathway through the alteration of tumor suppressor genes, such as *serine/threonine kinase 11 STK* (STK11), *TSC complex subunit 1* (*TSC1*), *TSC complex subunit 2* (*TSC2*) and *AKT serine/threonine kinase 1* (*AKT1*) [105,106,107].

Consistent with its cellular function, HIF1-α is often overexpressed in many tumors, including colorectal tumors [108,109,110,111,112].

In addition, the accumulation of reactive oxygen species (ROS), resulting from the ETC alterations due to the SDH disfunction, also contribute to the stabilization of the HIF1-α transcription factor by the oxidation of Fe^2+^ to Fe^3+^, since Fe^2+^ is an essential PHD co-factor [113]. Consequently, ROS accumulation contributes to the HIF1-α stabilization and to the activation of downstream genes that induce cell growth, cell migration and cell invasion, hence promoting EMT (Figure 2 and Figure 3) [114].

Current evidence also suggests that succinate can activate the vascular endothelial growth factor A (VEGFA) overexpression mediated by the mitogen-activated protein kinases 1 and 3 (MAPK1/ERK2 and MAPK3/ERK1) and the signal transducer and activator of transcription 3 (STAT3) signaling pathways through binding to the SUCNR1 receptors on the plasma membrane surface of endothelial cells [83,115]. In agreement with these observations, alterations in both ERK- and STAT3-mediated signaling pathways are associated with disease progression and aggressiveness in colorectal cancers [116,117,118,119,120,121,122].

## 7. Dietary and Environmental Factors which Could Influence SDH Activity and Succinate Concentration

The nutrients we consume through our diet play a crucial role in maintaining our health and preventing diseases, including cancer. Recent research has highlighted the inter-individual and organ-specific variability in nutrient effects, which may be influenced by genetic factors [123]. Precision nutrition therapeutics is an emerging concept that aims to optimize diet based on individual characteristics.

Studies have shown that diet can modulate succinate concentration in mice, with high-fat diets increasing succinate levels and fiber-rich diets counteracting this effect and reducing inflammation caused by dietary fat overload [124]. In addition, a recent study on C57BL/6J mice found that a western diet, characterized by high fat, glucose and cholesterol content, downregulated the expression of genes encoding for SDH subunits and SIRTUIN3, an SDH activator, and repressed SDH enzymatic activity, resulting in elevated liver succinate levels [125].

Food contamination with specific pesticides, such as those used against fungi, can also inhibit SDH activity. In vitro experiments using inhibitors such as nitropropionic acid, an irreversible inhibitor, or malonate, a competitive inhibitor, showed that an adverse effect of more than 30% could be observed in the inhibition of enzyme activity. Future studies are needed to clarify whether prolonged exposure to low doses of SDH inhibitor may be harmful for human and animal health [126].

## 8. Conclusions

Current research indicates that alterations in the SDH enzyme and in succinate concentration are key factors in the onset and progression of many cancers, including colorectal cancer, most likely due to the activation of the EMT process [61]. Succinate, therefore, appears to be a metabolite that promotes tumor progression and EMT activation through both intrinsic and extrinsic pathways.

Indeed, its effects are partly mediated by the specific succinate receptor, SUCNR1, which has also been recently shown to contribute to the early stages of inflammation in colonic epithelial cells [127]. Although several SUCNR1 receptor inhibitors have been synthesized to date [82], with compelling evidence that proves their efficacy in blocking tumor and metastatic progression in various cancer cell types [128,129], their clinical implication has been limited due to the serious side effects caused by SUCNR1 inhibition. In fact, SUCNR1 is involved in many fundamental physiological processes, such as thermogenesis [130]. 

In order to develop effective and safe therapies for patients with colorectal cancer, it is crucial to invest in the standardization of a precise drug that selectively targets SUCNR1 receptors in metastatic cells while sparing healthy cells. However, emerging evidence suggests that other alternative therapeutic targets, such as TRAP1, SDH, ERK1/2 and STAT3 and HIF1-α also deserve further investigation. This is particularly relevant for the management of specific subgroups of patients, such as GIST patients. Germline or epigenetic alterations of *SDH* are found in approximately 5.10% of GIST and are associated with particularly aggressive tumors that have an early onset, poor prognosis, and a poor response to conventional therapies compared to GIST with pathogenic variants in the *KIT proto-oncogene*, and *platelet-derived growth factor receptor alpha* (*PDGFRA*) genes [131]. Therefore, identifying and standardizing new precision therapies for the management of these patients is an urgent need.

## Figures and Tables

**Figure 1 biomedicines-11-01428-f001:**
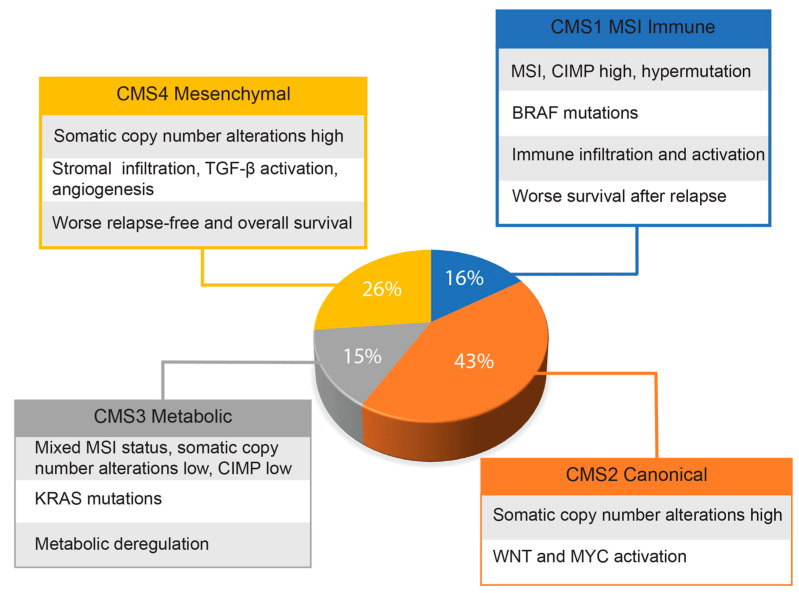
Approximate frequencies of each CRC molecular subtype. The main features of each molecular subtype are indicated.

**Figure 2 biomedicines-11-01428-f002:**
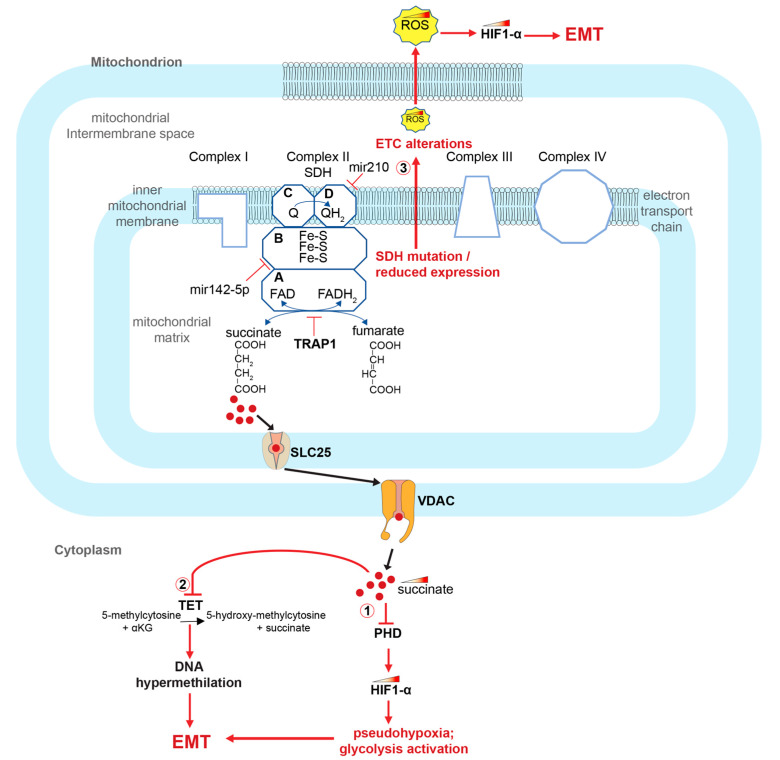
Role of SDH alterations and succinate accumulation in the activation of the EMT process in CRCs. SDH alterations cause succinate accumulation and activate EMT in CRCs through at least 3 ways: (1) inhibiting PHD and thereby stabilizing HIF1-α; (2) inhibiting the TET enzyme and thus generating DNA methylation alterations; (3) causing alterations of the ETC and accumulation of ROS, which, again, stabilizes HIF1-α. HIF1-α (hypoxia-inducible factor 1-alpha); TRAP1 (TNF receptor associated protein 1); SDH (succinate dehydrogenase); PHDs (prolyl hydroxylase domain enzymes); TET (ten-eleven translocation (TET) methylcytosine dioxygenases); ROS (reactive oxygen species); EMT (epithelial–mesenchymal transition); VDACs (voltage-dependent anion-selective channel proteins); SLC25 (solute carrier family 25 members).

**Figure 3 biomedicines-11-01428-f003:**
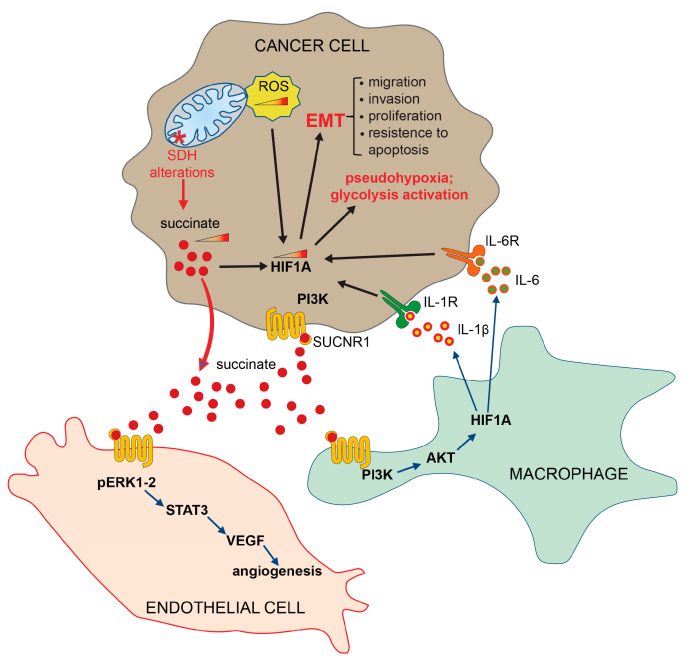
Extrinsic pathways activated by succinate. Succinate, secreted by tumor cells, binds its specific receptor SUCNR1 on target cells, such as endothelial cells, macrophages and other tumor cells, to induce EMT, angiogenesis and inflammatory pathways. HIF1-α (hypoxia-inducible factor 1 subunit alpha); SDH (succinate dehydrogenase); PI3Ks (phosphoinositide 3-kinases); AKT (protein kinase B, PKB); ERK (extracellular signal-regulated kinase); STAT3 (signal transducer and activator of transcription 3); VEGF (vascular endothelial growth factor); IL-1R (interleukin-1 receptor); IL-1β (interleukin-1 beta); IL-6 (interleukin-6); IL-6R (interleukin 6 receptor); SUCNR1 (succinate receptor 1); ROS (reactive oxygen species); EMT (epithelial–mesenchymal transition).

**Table 1 biomedicines-11-01428-t001:** Hereditary cancer syndromes.

SYNDROMES	INVOLVED GENES
** *Hereditary non-polyposis colorectal cancer* **	
LYNCH	*MSH2*, *MLH1*, *MSH6*, *MSH3*, *PMS2*, *EPCAM*
NONPOLYPOSIS CRC-MSS	*RPS20*
** *Familial adenomatous polyposis syndromes* **	
FAP/AFA: familial adenomatous polyposis (including Gardner syndrome and Turcot Syndrome)/attenuated-FAP	*APC*
PPAP: polymerase proofreading-associated polyposis	*POLE*, *POLD1*
MAP: MUTYH associated polyposis	*MUTYH*
NAP: NTHL1-associated polyposis	*NTHL1*
MSH3 polyposis	*MSH3*
** *Hamartomatous polyposis syndromes* **	
PJS: Peuts–Jeghers syndrome	*STK11*
PHTS: PTEN hamartoma tumor syndrome (including Cowden syndrome and Bannayan-Riley-Ruvalcaba syndrome)	*PTEN*, *SDHD*
JPS: juvenile polyposis syndrome	*BMPR1A*, *SMAD4*
** *Mixed polyposis* **	
HMPS: hereditary mixed polyposis syndrome	*GREM1*, *BRAF*
** *Serrated adenomas* **	
SPS: serrated polyposis syndrome	*RNF43*

## Data Availability

Not applicable.

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
