# Peer review of "The Epithelial to Mesenchymal Transition in Colorectal Cancer Progression: The Emerging Role of Succinate Dehydrogenase Alterations and Succinate Accumulation"

_biomedicines, 2023, doi:10.3390/biomedicines11051428_

Round 1

Reviewer 1 Report

This paper covers very interesting topics of EMT. Contents are generally superb although some points are missing. After improvements in the contents, this paper can be very useful. Specific comments are below.

It would be very useful to discuss important missing factors. In a new short section, the author can discuss environmental, dietary, and life habit factors which influence pathogenic processes. How those modify the TCA cycle, gene mutations, and succinate levels may be interesting. These factors may influence molecular pathology in each patient.

In bigger perspectives, research on life style, foods, drinks, and other habitual and environmental factors should be used with analyses of personalized molecular biomarkers. The authors can discuss molecular pathological epidemiology research that can investigate those factors, clinical consequences and molecular pathology, together. Molecular pathological epidemiology has been elaborated previously; eg, Ann Rev Pathol 2019, J Gastro 2017, Curr Colorectal Cancer Rep 2017, etc. and can be a promising direction to investigate EMT and tumor progression.  

Author Response

Point 1: It would be very useful to discuss important missing factors. In a new short section, the author can discuss environmental, dietary, and life habit factors which influence pathogenic processes. How those modify the TCA cycle, gene mutations, and succinate levels may be interesting. These factors may influence molecular pathology in each patient. 

In bigger perspectives, research on life style, foods, drinks, and other habitual and environmental factors should be used with analyses of personalized molecular biomarkers. The authors can discuss molecular pathological epidemiology research that can investigate those factors, clinical consequences and molecular pathology, together. Molecular pathological epidemiology has been elaborated previously; eg, Ann Rev Pathol 2019, J Gastro 2017, Curr Colorectal Cancer Rep 2017, etc. and can be a promising direction to investigate EMT and tumor progression.  

Response 1: We appreciate the insightful feedback provided on our manuscript. We agree that these highlighted factors are critical in pathogenic processes and have added a new section to the manuscript to address this topic, entitled: “7. Dietary and environmental factors which could influence SDH activity and succinate concentration”, page 10-11, lines 360-379. We have included references to relevant studies to support this discussion.

We have carefully considered Reviewer 1's comments and believe that the improvements made to the manuscript have strengthened it significantly. We appreciate the reviewer's valuable feedback and hope that these revisions have addressed his concerns.

Sincerely,

Marina De Rosa

Reviewer 2 Report

Authors should provide more clinical information and potential application of therapies/treatments in this manuscript.

Author Response

Response to Reviewer 2 Comments

Point 1:

Authors should provide more clinical information and potential application of therapies/treatments in this manuscript.

Response 1:  We fully acknowledge the critical need for developing safe and effective treatments for patients, and in response to this feedback, we have expanded this topic in the discussion section of our revised manuscript (page 11; lines 398-405).

We hope that these revisions address the concerns raised by Reviewer 2 and improve the overall quality of our manuscript.

Sincerely,

Marina De Rosa